# Direct observation of strong surface reconstruction in partially reduced nickelate films

Chao Yang [1] ✉, Rebecca Pons[1], Wilfried Sigle[1], Hongguang Wang [1],
Eva Benckiser [1], Gennady Logvenov[1], Bernhard Keimer [1] &
Peter A. van Aken [1]

The polarity of a surface can affect the electronic and structural properties of oxide thin films through electrostatic effects. Understanding the mechanism behind these effects requires knowledge of the atomic structure and electrostatic characteristics at the surface. In this study, we use annular bright-field imaging to investigate the surface structure of a $Pr_{0.8}Sr_{0.2}NiO_{2+x}$ ($0 < x < 1$) film. We observe a polar distortion coupled with octahedral rotations in a fully oxidized $Pr_{0.8}Sr_{0.2}NiO_3$ sample, and a stronger polar distortion in a partially reduced sample. Its spatial depth extent is about three unit cells from the surface. Additionally, we use four-dimensional scanning transmission electron microscopy (4D-STEM) to directly image the local atomic electric field surrounding Ni atoms near the surface and discover distinct valence variations of Ni atoms, which are confirmed by atomic-resolution electron energy-loss spectroscopy (EELS). Our results suggest that the strong surface reconstruction in the reduced sample is closely related to the formation of oxygen vacancies from topochemical reduction. These findings provide insights into the understanding and evolution of surface polarity at the atomic level.

Polarity at interfaces and surfaces in complex oxide thin films plays a critical role in their physical and chemical properties such as ferroelectricity[1–7], superconductivity[8], magnetism[9], and catalysis[10]. By controlling the crystal plane orientation and termination during film growth, polar surfaces, and interfaces can be obtained, where structural distortions form due to the interaction of short-range covalency and long-range electrostatic effects[11–14]. At surfaces, the abruptly reduced coordination significantly alters the lattice and electronic structures, potentially affecting the overall physical and chemical properties of the thin oxide material. For example, the polar distortion and octahedral rotations at the polar surface of a $LaNiO_3$ single-crystal film weaken the hybridization of Ni $3d$ and O $2p$ orbitals[13], resulting in a decreased metallicity and a thickness-dependent transport behavior[15–17]. Capping with an insulating $LaAlO_3$ layer[17] or changing the surface termination[13] can reduce or eliminate the polar distortion, and thus recover the metallic conductivity of the $LaNiO_3$ film again.

Furthermore, the excess charges on the polar surfaces lead to the emergence of different electronic states such as charge-density waves, localized electron polarons[18], and two-dimensional electron gas (2DEG)[19]. Additionally, surface polarization can effectively boost electro- and photocatalytic performance by tuning the adsorption intensity and charge separation at catalyst surfaces[10]. Therefore, exploring and controlling surface polarity is essential for engineering functionalities of electronic devices and surface catalysts.

Exploiting and controlling surface polarity in complex oxide thin films is a challenging task, due to the difficulty in growing precisely controlled single atomic layer terminated crystal film surfaces and probing their local atomic and electronic structures[5]. However, theoretical studies have demonstrated that electronic orbitals can be modified by polar surface distortions. For example, the $NiO_2$-terminated negatively charged surface in $LaNiO_3$ thin films has been found to possess a large orbital polarization due to the eliminated Ni-O bond

[1]Max Planck Institute for Solid State Research, Stuttgart, Germany. ✉e-mail: c.yang@fkf.mpg.de

in out-of-plane direction, which is a promising approach to mimic the electronic configuration of high-temperature cuprate superconductors[20,21]. The polar distortion at the $NiO_2$-terminated negatively charged surface elongates the out-of-plane Ni-O bond, which lowers the orbital energy of $d_{z^2}$ and further enhances the orbital polarization. However, the study of surface modification for orbital engineering purposes is still lacking due to the difficulty of controlling the surface polarity in practice. Additionally, the surface termination can influence polar distortions in $LaNiO_3$ thin films, determined from the analysis crystal truncation rods (CTR) measured by synchrotron x-ray diffraction, thereby altering the electronic conduction[13]. Other experimental observations have revealed different phases at polar surfaces, such as polarization-controlled surface reconstruction in a $Pb(Zr_{0.2}Ti_{0.8})O_3$ film[11] and competing electronic states, e.g., charge-density waves and localized electron polarons, at a $TaO_2$-terminated polar surface in a $KTaO_3$ film[18].

The infinite-layer structure of nickelates, which upon appropriate cation doping becomes superconducting[22–26], can only be synthesized through oxygen deintercalation via topochemical reduction. In the latter process, the compositional and structural changes can lead to modifications in polarity at interfaces or surfaces due to the removal of the apical oxygen ions[27–29]. Polarity at a perfect $NdNiO_2/SrTiO_3$ interface induces 2DEG formation due to strong occupation of interfacial Ti according to theoretical calculations[27], which is similar to the $LaAlO_3/SrTiO_3$ superconducting system[30]. However, the experimental results

show that the residual oxygen, elemental mixing, atomic reconstruction, and Ni valence modulations can compensate for the polar discontinuity induced by the interface polarity at the $NdNiO_2/SrTiO_3$ interface[14]. In terms of surface polarity, theoretical calculations predict a high Ni $e_g$ orbital polarization of 35% due to the elimination of out-of-plane Ni-O bonding associated with polar distortion[27]. However, the experimental study of the effects of the surface property is lacking due to the difficulty in sample synthesis and characterization. Recently, a charge-density wave state has been observed in the infinite layer (R, Sr) $NiO_2$ (R = La, Nd)[31], where the presence of a $SrTiO_3$ top layer can lead to controversial results about the charge density wave state[32,33], possibly indicating the potentially critical influence of polarity on the surface or interface electronic state. By using a $SrTiO_3$ capping layer, the polarity at the surface is greatly reduced due to elemental mixing and residual oxygen. In addition, the homogeneity of oxygen deintercalation in the bulk region can also affect the geometry of the infinite layer phase. Understanding the reduction process is crucial for better synthesis of the infinite layer phase. Therefore, detailed studies of interface, surface, and geometry effects at the atomic scale are essential to understand, exploit, and control oxygen deintercalation in nickelate superconductors and similar systems. In this study, we aimed to control the extent of oxygen deintercalation to modify the surface polarity in a $Pr_{0.8}Sr_{0.2}NiO_{2+x}$ film and used a combination of atomic-resolution electrostatic-field imaging via a combination of 4D-STEM and STEM-EELS to directly image the variation of atomic and electronic structures

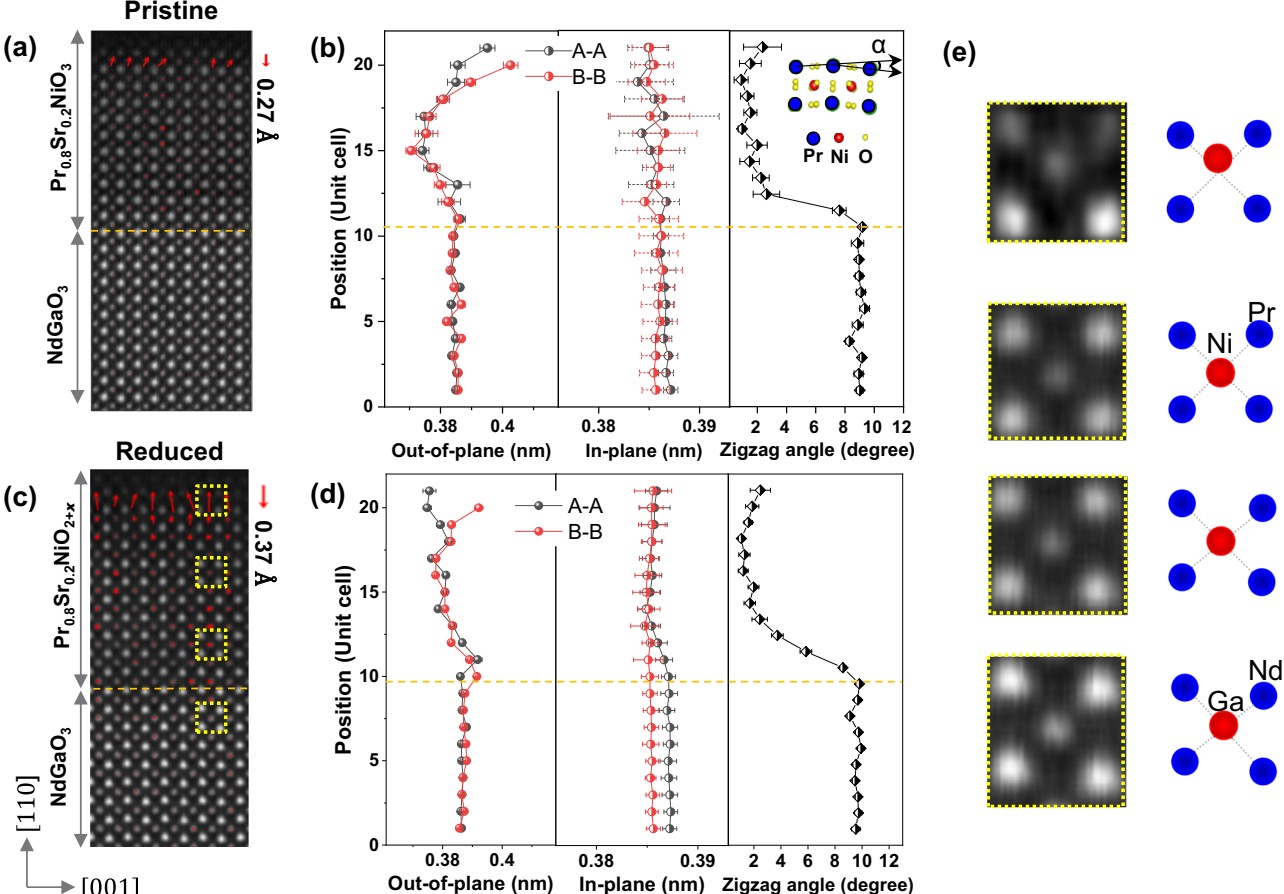

**Fig. 1 | Atomic structures in pristine $Pr_{0.8}Sr_{0.2}NiO_3$ and topochemical reduced $Pr_{0.8}Sr_{0.2}NiO_{2+x}$ films.** HAADF images and the corresponding Ni displacement maps at the surfaces of **a** $Pr_{0.8}Sr_{0.2}NiO_3$ and **c** $Pr_{0.8}Sr_{0.2}NiO_{2+x}$ films. Variations of lattice distances (in out-of-plane and in-plane lattice directions) and A–A–A (A: Nd, and Pr) zigzag angle in **b** $Pr_{0.8}Sr_{0.2}NiO_3$ and **d** $Pr_{0.8}Sr_{0.2}NiO_{2+x}$ films. The yellow dashed lines indicate the film/substrate interfaces. The α indicates zigzag angle.

**e** The enlarged HAADF images from the yellow dashed boxes in **c** and the corresponding schematic structures for the evolution of atomic structure from the $NdGaO_3$ substrate to the film surface. Red spots represent the atom columns of Ni or Ga, and blue spots are Nd or Pr atom columns. The error bar is calculated by averaging multiple unit cells on each row.

at the surface layer. Our results provide direct evidence of the homogenous Sr doping in the $Pr_{0.8}Sr_{0.2}NiO_3$ film, and how the polar distortion and oxygen octahedra rotation in the surface layer are connected. A distinct and even stronger surface reconstruction is found in the $Pr_{0.8}Sr_{0.2}NiO_{2+x}$ film, synthesized via topochemical reduction from the same piece of the $Pr_{0.8}Sr_{0.2}NiO_3$ sample, which is associated with a thickness-dependent oxygen deintercalation.

## Results and discussion

Figure 1 illustrates the use of STEM imaging to study the polar surfaces of $Pr_{0.8}Sr_{0.2}NiO_3$ and $Pr_{0.8}Sr_{0.2}NiO_{2+x}$ films. The cations in these films (Pr, Nd, Ga, and Ni) can be easily distinguished in the HAADF images due to their varying atomic numbers. In Fig. 1a, the displacement map of the B atoms (B: Ni and Ga) shows a Ni displacement of ~0.27 Å at the surface layer, which gradually decreases to zero within ~3 unit cells below the surface. This displacement is consistent with previous studies on $LaNiO_3$ films using CTR[13,34]. In the topotactically reduced $Pr_{0.8}Sr_{0.2}NiO_{2+x}$ films, we observe an increase in the magnitude of the Ni displacement to ~0.37 Å for a reduction time of 6 h (Fig. 1c) and to ~0.45 Å for a reduction time of 18 h (Figure S1). Figure S2 shows the overview HAADF images of the pristine and reduced samples, where there are no apparent defects in the inner layer of these films. To quantify these structural changes, we use Gaussian fitting and center-of-mass refinement techniques based on the Atomap python package[35]. We define the atomic positions from the HAADF images and quantify the structural parameters, including in-plane and out-of-plane lattice distances, and the A–A–A (A: Pr and Nd) zigzag angle as depicted in Fig. 1b, d. The difference between the A–A and B–B out-of-plane lattice spacings shows the Ni displacement. There is no visible change in the B–B in-plane lattice distance, suggesting a tensile strain in the $Pr_{0.8}Sr_{0.2}NiO_3$ film. The A–A in-plane lattice distance has a small decrease from the substrate to the film due to the larger zigzag angle of Nd–Nd in bulk $NdGaO_3$ compared to that of Pr–Pr in bulk $PrNiO_3$. Another possible reason is that adjusting the zigzag angle can release the tensile strain to some extent, as the decrease of the zigzag angle is apparent in Fig. 1b, d. Enlarged HAADF images and corresponding schematic structural models in Fig. 1e clearly demonstrate the evolution of atomic structures from a zigzag structure at the substrate interface to a polar distortion at the film surface.

To determine the polar structure at the surface layer, we identified oxygen atomic columns and quantified the relative displacement of the oxygen and B cations at the $BO_2$ plane. Figure 2a, b shows the experimental high-resolution annular bright-field (ABF) images at the

surface regions in $Pr_{0.8}Sr_{0.2}NiO_3$ and $Pr_{0.8}Sr_{0.2}NiO_{2+x}$ films, respectively. The figures reveal the coexistence of oxygen octahedra rotation and polar distortion at the subsurface of the pristine sample, consistent with that in $LaNiO_3$ film measured by synchrotron x-ray surface diffraction[34]. The polar distortion dominates the structural changes in the surface region. Notably, a stronger polar distortion occurs at the subsurface of the reduced $Pr_{0.8}Sr_{0.2}NiO_{2+x}$ film, and nonpolar rotation is not detected, as shown in Fig. 2c. The enlarged ABF images and corresponding schematic structures in Fig. 2d highlight the variations of oxygen octahedra. The terminating $NiO_2$ layer shows an obvious buckling of Ni–O–Ni and there is a strong polarization in the adjacent unit cell. Figure 2e quantitatively compares the relative displacements of Ni and O columns at the $NiO_2$ plane in the out-of-plane direction between the pristine and reduced samples. The largest polar displacement is ~0.32 Å in the pristine sample and ~0.56 Å in the reduced one, indicating a significantly enhanced polar state at the surface region of the $Pr_{0.8}Sr_{0.2}NiO_{2+x}$ film. Such a large polar distortion strongly affects electronic transport by modifying orbital overlap between Ni 3$d$ and O 2$p$, possibly inducing competation between local structures at the surface, interior, and interface and modifying the conductivity of the entire film[34]. The polarization decays within ~3 unit cells in both pristine and reduced samples, in agreement with the observed decay of polarization in a $LaNiO_3$ film[34].

In order to understand the mechanism of electrostatic screening of the charged surface, we performed 4D-STEM experiments to measure atomic-scale electrostatic fields and charge distribution at the surface of a reduced sample with a strong polar state. To minimize scanning distortion and improve the signal-to-noise ratio, we optimized the reconstructed images in Fig. 3 by aligning and summing four consecutive frames. We carefully prepared and measured the sample, which has a thickness of around 5.2 nm, determined from position-averaged convergent beam electron diffraction (PACBED) patterns (see Figure S3b–e), to ensure a quantitative understanding of the atomic electric field[36,37]. We also identified no tilt in the sample, as confirmed by PACBED patterns and by the oxygen octahedral structure from the ABF image in Fig. 2. Figure 3b shows the modulus of the atomic electric field map. The electric fields surrounding Pr and Nd atoms are stronger than those of Ni and Ga atoms due to their larger atomic numbers, and exhibit approximately a rotationally symmetric distribution and similar magnitude (see Figure S4). Notably, the magnitude and symmetry of the fields surrounding Ni atom columns show a pronounced change near the surface, as seen in Fig. 3d. We extracted the line profile of the field strength surrounding B (Ni and Ga) atom

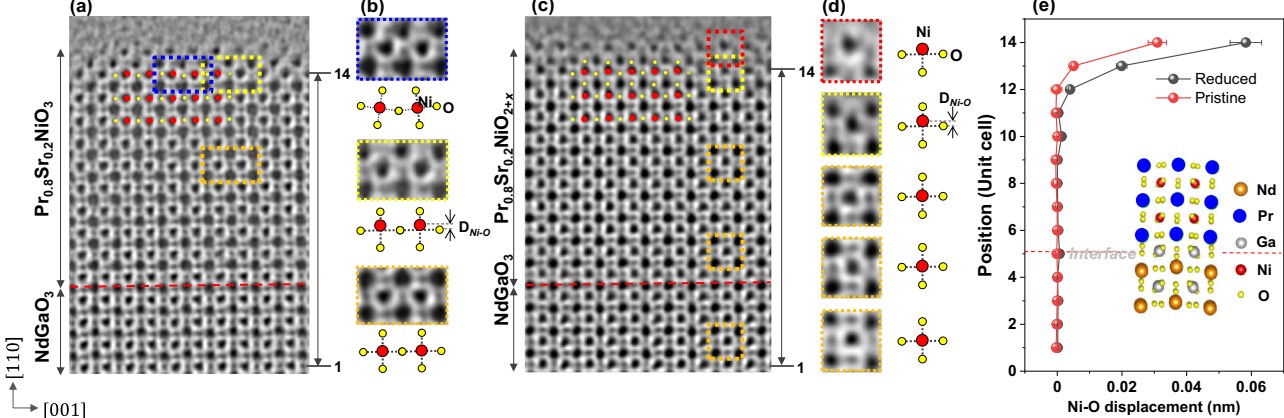

**Fig. 2 | Oxygen sublattices in $Pr_{0.8}Sr_{0.2}NiO_3$ and $Pr_{0.8}Sr_{0.2}NiO_{2+x}$ films.** ABF images of **a** $Pr_{0.8}Sr_{0.2}NiO_3$ and **c** $Pr_{0.8}Sr_{0.2}NiO_{2+x}$ films. **b** Enlarged ABF images and the corresponding schematic structural models (below) show the variation of oxygen octahedra in the pristine sample. $D_{Ni-O}$ indicates the relative displacement of Ni and O. **d** Enlarged ABF images and the corresponding schematic structural

models (right) show the variation of oxygen octahedra in the reduced sample. **e** The relative displacement of Ni and O at the $NiO_2$ plane in the out-of-plane direction. The schematic atomic structure at the $PrNiO_3/NdGaO_3$ interface. The error bar is calculated by averaging multiple unit cells on each row.

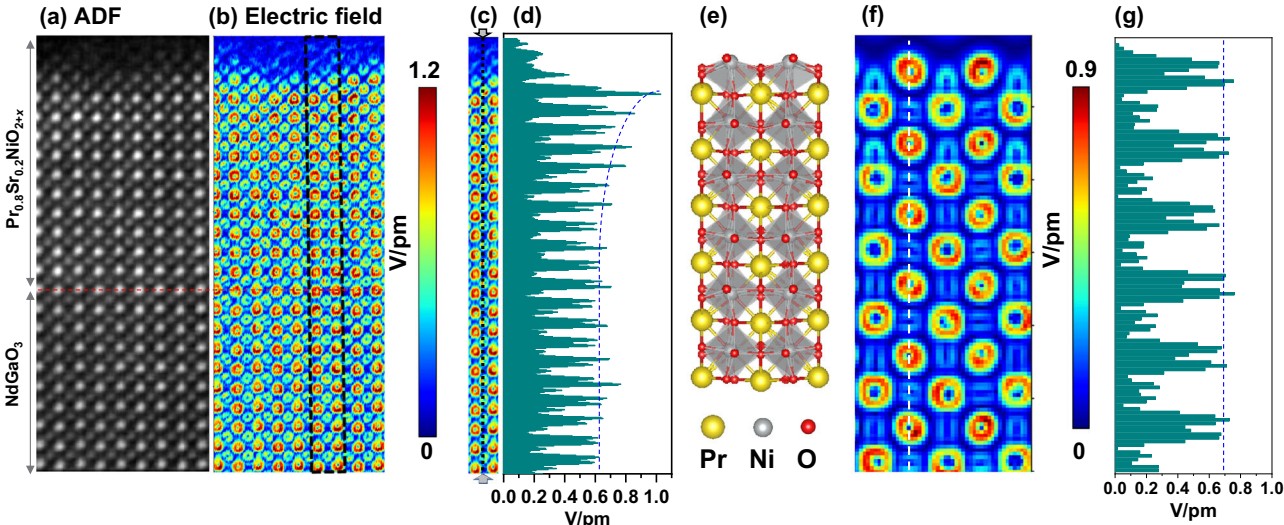

**Fig. 3 | Extracted information from a 4D-data set of the reduced Pr$_{0.8}$Sr$_{0.2}$NiO$_{2+x}$ film.** Reconstructed atomic-column-resolved **a** ADF, and **b** electric field images. **c** The electric-field map extracted from the region marked with a black dashed box in **b**. **d** The line profile extracted from the region marked with a black dashed line in **c** indicates the changes in the magnitude of the electric field around B (B: Ni, and Ga) atoms. **e** A PrNiO$_3$ supercell created from the ABF image of a reduced sample with polar distortions at the upper surface. **f** Simulated electric field map. **g** The corresponding line profile of the electric field map marked with a white dashed line in **f**.

columns marked with a black dashed line in Fig. 3c, showing that the magnitude becomes larger in several unit cells near the surface. What is the relationship between polar distortions and electric field variations near the surface? To clarify this, we performed electric field simulations with different sample thicknesses as shown in Fig. 3e–g and Figure S3f–k. We created a supercell for 4D-STEM simulations without considering Sr doping as shown in Fig. 3e, but including polar distortions in the surface region, corresponding to the atomic structure in the ABF image of the reduced sample. Figure 3f shows the simulated electric field map for a sample thickness of 5.2 nm. There is no clear difference in the strength of the atomic electric field surrounding Ni atoms according to the line profile in Fig. 3g.

In addition, we substituted oxygen (O) and nickel (Ni) columns with fluorine (F) and copper (Cu) atoms in the supercell setup for our 4D-STEM simulations. This swap allowed us to probe how subtle shifts in atomic potential influence the electric field, as visually depicted in Figure S5. Notably, we observed a lack of significant differences in the strength of the electric field around neighboring O and F columns, as well as Ni and Cu columns. This observation prompts us to consider the possibility that the effect of lattice vibrations might mask the field changes resulting from minor shifts in atomic potentials within this system. This suggests that the field variations in our experimental results mainly stem from valence change or charge redistribution, rather than from diffraction contrast induced by structural changes. The strong polar state in the reduced sample could induce electronic and atomic reconstructions, and oxygen vacancies can modify the surface polarization and Ni valence. Therefore, the observed stronger electric field surrounding Ni atoms at the surface region may mainly result from the change of charge distribution around Ni columns. Additionally, we compared the simulated electric field maps for different sample thicknesses in Figure S3h–k. In samples with thicknesses of 8.3 nm and 9.9 nm, the reconstruction of the electric field from momentum transfer is no longer reliable due to the artifacts of diffraction contrast caused by beam broadening. The rotationally symmetric distribution of the atomic electric field surrounding atoms is broken. It is worth noting that the field strength of Ni atoms decreases from ~0.7 V/pm to 0.23 V/pm with increased sample thickness from 5.2 nm to 9.9 nm. This magnitude of the atomic electric field surrounding Ni atoms from the simulation is consistent with the value

(~0.7 V/pm, see Fig. 3c) of the experimental result at the region far from the surface, further supporting the estimated sample thickness of around 5 nm measured by the PACBED patterns.

Next, we investigate the electronic structure of the reduced Pr$_{0.8}$Sr$_{0.2}$NiO$_{2+x}$ film using EELS fine-structure analysis of Ni-$L_{2,3}$ and O-$K$ edges near the surface. Figure 4(a) shows the ADF image for the EELS data analysis, with spectra extracted layer by layer from the regions marked with red dashed boxes. Layers 1-12 belong to the film. The extracted O-$K$ edge spectra in Fig. 4b identify the film, with a pronounced pre-peak indicating hybridization of O-$2p$ and Ni-$3d$ orbitals in the NiO$_6$ octahedra, which is sensitive to changes in Ni valence and Ni-O bond length. Figure 4c shows a gradual decrease in the maximum intensity ratio of peaks A and B of the O-$K$ edge, indicating a decrease in the hybridization of Ni $3d$ and O $2p$[13] or the formation of oxygen vacancy[38]. This is related to the increased resistivity of the reduced sample as shown in Figure S6. Figure 4d shows a shift of the Ni-$L_3$ edge to lower energies, suggesting a change in Ni valence. The analysis of the white-line ratio of Ni-$L_{2,3}$ edges in Fig. 4e reveals that the Ni valence decreases from the inner layer to the surface of the film, where the references of Ni$^{3+}$ and Ni$^{2+}$ are acquired from NdNiO$_3$[39] and NiO films[40], respectively. This is consistent with changes in the atomic electric fields surrounding Ni atom columns measured by 4D-STEM. In contrast, the Ni valence almost remains around Ni$^{3+}$ apart from a little decrease at the last unit cell in the pristine sample (see Fig. S7), resulting from a possible oxygen vacancy at the surface for compensating the surface charge. However, it is worth noting that except for the electronic structural reconstruction induced by polar distortion, an oxygen vacancy is another main factor to influence both Ni valences and the polar state.

To gain insight into the distribution of oxygen, we analyzed the elemental distribution from O-$K$ EELS maps that were resolved layer-by-layer (see Fig. 5). We identified the cation atom columns in both pristine and reduced samples by using the Ga-$L_{2,3}$, Nd-$M_{4,5}$, Pr-$M_{4,5}$, and Ni-$L_{2,3}$ edges. The composite map in Fig. 5b, e shows the elemental distribution from the substrate to the film surface. Due to the high-energy loss of the Sr-$L_{2,3}$ edge and low Sr doping concentration, the Sr signal was barely measurable using STEM-EELS. Therefore, we employed atomic-resolution STEM-EDX, which revealed that the Sr signal was homogenous throughout the film (Figure S8). We

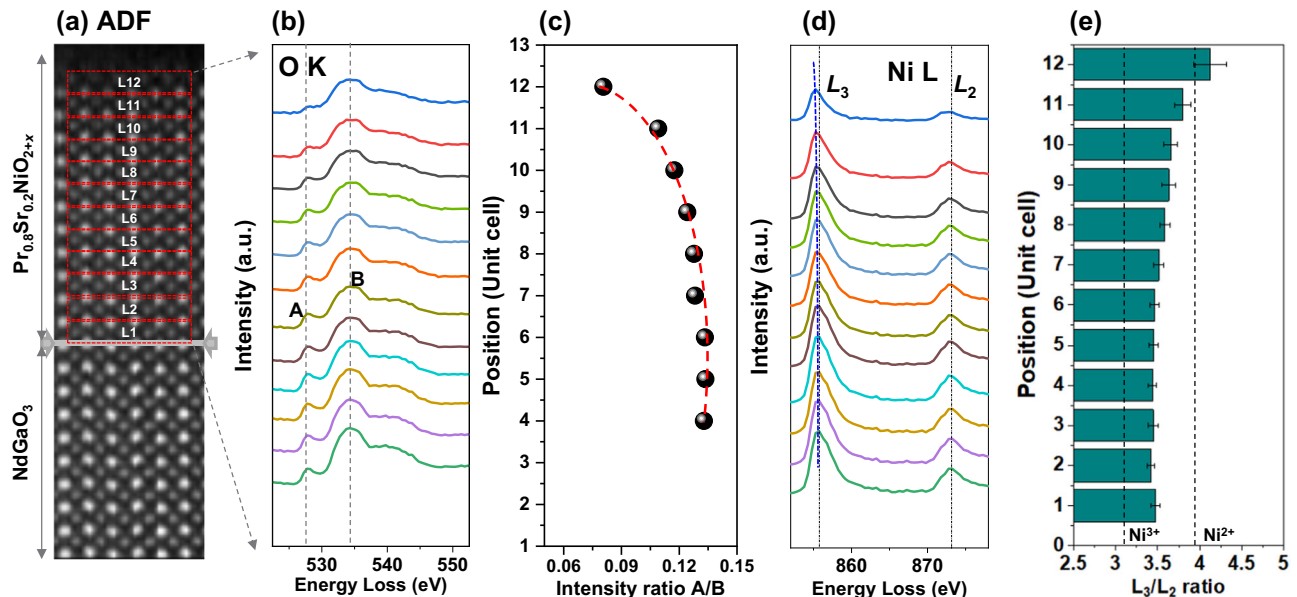

**Fig. 4 | EELS measurements of the Ni-$L_{2,3}$ and O-$K$ edges in the reduced Pr$_{0.8}$Sr$_{0.2}$NiO$_{2+x}$ film. a** ADF image for EELS data analyses. **b** O-K edges and **d** Ni-$L_{2,3}$ edges extracted from the regions marked in **a**. **c** Calculated intensity ratios of peaks A and B from O-$K$ edges. **e** Ni-$L_{2,3}$ white-line ratios extracted from the regions marked in **a**. The error bar is calculated by averaging multiple regions on each row.

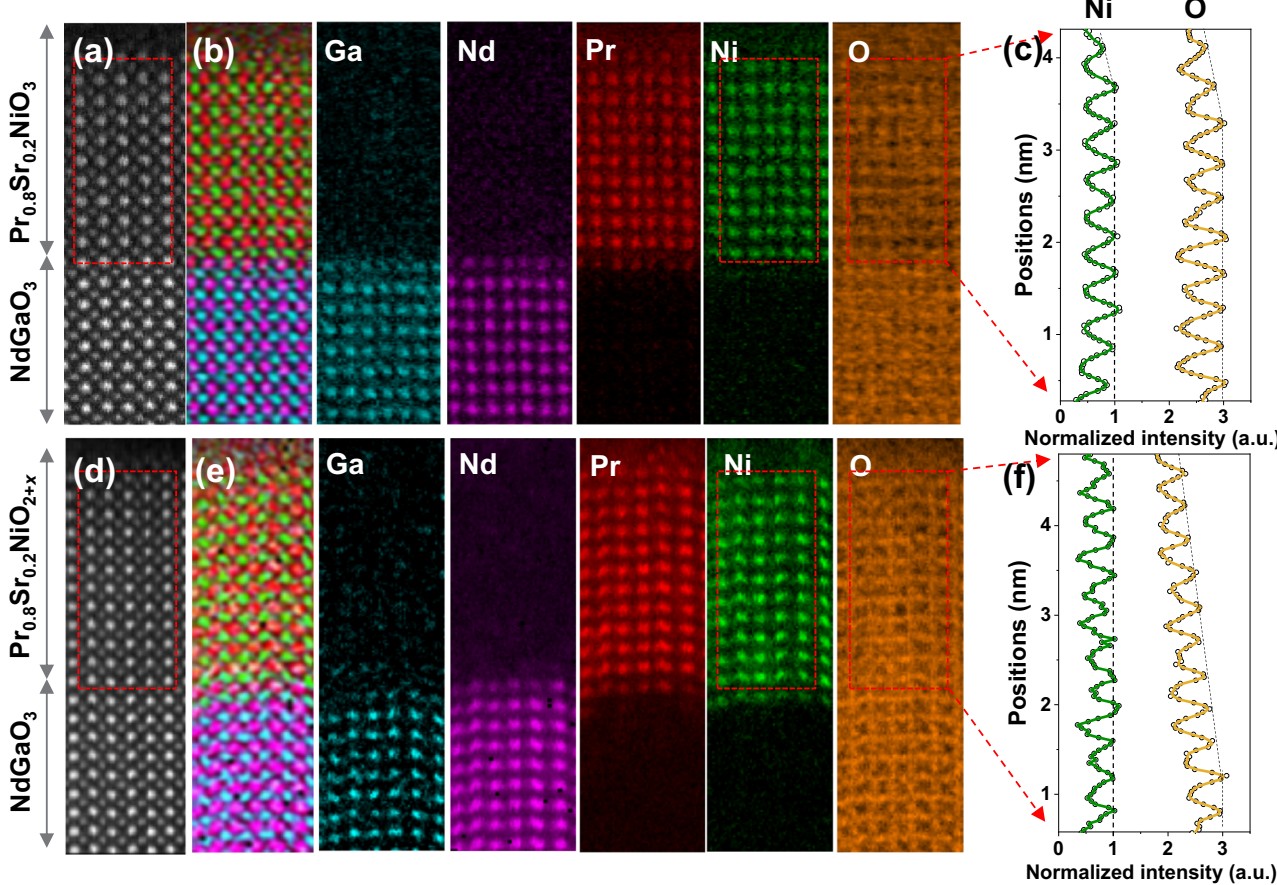

**Fig. 5 | Elemental distribution and variation of oxygen concentration in Pr$_{0.8}$Sr$_{0.2}$NiO$_3$ and Pr$_{0.8}$Sr$_{0.2}$NiO$_{2+x}$ films.** HAADF images of **a** Pr$_{0.8}$Sr$_{0.2}$NiO$_3$ and **d** Pr$_{0.8}$Sr$_{0.2}$NiO$_{2+x}$ films for EELS analyses. EELS maps of Ga, Nd, Pr, Ni, and O. **b**, **e** Color-coded mapping of Ga (blue), Nd (purple), Pr (red), Ni(green), and O (yellow). The normalized signal intensity of Ni and O in **c** Pr$_{0.8}$Sr$_{0.2}$NiO$_3$ and **f** Pr$_{0.8}$Sr$_{0.2}$NiO$_{2+x}$ films.

determined the oxygen content by integrating the O $K$ edges and normalizing it with Ni-$L_{2,3}$ edges in Fig. 5c, f. In the pristine sample (Fig. 5c), we observed a decrease in both Ni and O contents within one unit cell at the surface. Since the stoichiometry was still maintained, this was likely a result of specimen thickness. In contrast, the reduced sample had a distinct decrease in O signal intensity, with an estimated O/Ni ratio of ~2.5 at the surface. The Ni signal remained roughly the same throughout the area, consistent with the change in Ni valence mentioned previously. Additionally, the gradual decrease in oxygen concentration suggests a thickness-dependent deintercalation of oxygen.

The presence of surface polarity can cause structural distortion, electronic reconstruction, and charge redistribution to screen the bound charges at the surface[11]. In the pristine sample, we observed cooperative coupling of polar distortion and octahedral rotation underneath the negatively charged $NiO_2$ surface, which lacks direct atomic resolution observation in similar systems such as the $NiO_2$-terminated surface in a $LaNiO_3$ film[34] and a $PbTiO_3$ ferroelectric film[41]. The decay of polarization occurred within around three unit cells and there was no apparent oxygen deficiency to help screen the surface field. Polar distortion and octahedral rotation can strongly affect the electronic transport by modifying the overlap of Ni $3d$–O $2p$ orbitals. Quantification of the Ni–O–Ni bond angle enables to evaluate the $p$-$d$ orbital overlap[34]. The octahedral distortions occured within several unit cell at surfaces can lead to a thickness-dependent electronic modification, which plays an important role in tuning the electronic behavior[34]. According to a canonical tight binding approach, the changes in bond overlap can be estimated by a transfer integral $t_{pd}$ between Ni $3d$ and O $2p$ orbitals[34,42],

$$t_{pd} = k \frac{\cos(90 - \theta/2)}{d^{3.5}} \qquad (1)$$

where k is a constant determined by the $p$-$d$ orbital covalent hopping integral, $\theta$ is the in-plane Ni-O-Ni bond angle, and $d$ is the Ni-O bond length. The distortion in our sample reduces the Ni–O–Ni angle in the surface region, which reduces the overlap or hybridization between the Ni$3d_{x^2-y^2}$ and O $2p$ orbitals and leads to a reduction in the valence bandwidth[34,43,44]. The reduced bandwidth can lead to the opening of the charge transfer gap between the Ni $e_g$ and O $p$ valence bands, which can affect the electronic transport behavior[34]. Furthermore, DFT calculations were performed to study the effects of the polar distortion at the nickelate surface region on the density of states (DOS) of the O $2p$ and Ni $3d$ orbitals, as shown in Figure S9. For a negatively charged $NiO_2$ terminated surface, a Ni–O–Ni buckling structure is formed and the apical oxygen moves away from the Ni atoms due to the electrostatic field at the surface. Comparing the projected DOS of the distorted and undistorted surface structures, there is a decrease in electronic states near the Fermi level at the polar distorted surface. By introducing oxygen vacancies at the apical site, an infinite layer structure is formed, which is accompanied by a reduction of the out-of-plane Ni-Ni distance and a surface polar distortion after structure optimization. The corresponding electronic states of Ni $3d$ and O $2p$ near the Fermi level are further reduced, indicating a reduction in the charge transfer capability between Ni and oxygen.

Notably, the layer-selective topotactic reduction modifies the oxygen structures in nickelates in an interesting way. In the reduced sample, stronger polar distortion formed at the surface due to oxygen deintercalation. The polarization similarly decayed within around three unit cells. Interestingly, the thickness-dependent oxygen deintercalation resulted in a gradually decreased oxygen concentration from the substrate/film interface to the surface. The removal of apical oxygen from $NiO_6$ octahedra is easier than basal oxygen[45], leading to an asymmetric distribution of missing oxygen in the PrO layers in the out-of-plane direction, which strengthens the electrostatic field from

the layer-by-layer structure perspective. In principle, the polar discontinuity due to the charge imbalance at the polar interface or surface leads to electronic and atomic reconstructions to avoid the polar instability[30]. For example, according to the theoretical calculation, the electronic reconstruction at the $NdNiO_2/SrTiO_3$ interface can partially screen the built-in electrostatic field by charge transfer between Ni sites in adjacent layers, but retain some residual electrostatic energy and lead to atomic reconstruction[29]. There is a ~ 0.2 Å displacement of Ni atoms in the $NiO_2$ layer at the $NdNiO_2/SrTiO_3$ interface[29]. This is similar to our previous experimental results measured by STEM at the interfaces in the $8NdNiO_2/4SrTiO_3$ superlattice, although residual oxygen is present at the interface[14]. In the case of the partially reduced sample, the electrostatic effect is particularly pronounced in the unit cells near the surface, where the polarity is significantly enhanced. The stronger polar distortion can result in a large orbital polarization due to the missing apical oxygen or elongation of Ni-O bond. The oxygen vacancies in the reduced sample also alter the valence of Ni, as evidenced by atomic electric field mapping and white line ratio calculations of Ni-$L_{2,3}$ edges. These maps reveal an enhanced field strength surrounding Ni atom columns near the surface when compared to the film's inner layer and substrate. This enhancement is primarily related to the increased valence charge of the Ni, as determined by EELS. Electric field simulations of the stoichiometric phase $PrNiO_3$ also show comparable field magnitudes with the experimental values for the film inner layer. The real-space imaging of atomic electric fields thus provides direct information for measuring charge variations. The lower formation energy of oxygen vacancies at the surface of perovskite oxide[11] also allows for the formation of oxygen vacancies at the $NiO_2$ surface layer, which can compensate for the surface electrostatic field due to its asymmetric bonding. Other factors such as absorption of atoms and faceting at the surface may also contribute to depolarization. However, obvious surface reconstructions are observed to screen the polar field and has been systematically analyzed at atomic scale. A secondary phase may form under a stronger surface polarity in this system. This could be a reason for the instability of the infinite layer phase that cannot be synthesized directly.

In summary, our study provides real-space imaging of structural distortion and electronic reconstruction induced by electrostatic effects at negatively charged polar surfaces. The surface reconstruction varies in the pristine and partially reduced samples. Octahedral rotations and polar distortion coexist in the pristine sample, while stronger polar distortion occurs in the partially reduced samples, which is closely related to the presence of oxygen vacancies. The screening lengths of the depolarization field in both pristine and reduced samples are around three unit cells. The atomic electric-field-mapping method allows us to directly image the evolution of electronic structure, and we observed a gradual increase in field strength surrounding Ni atoms, indicating corresponding changes in the Ni valence state. This is consistent with our EELS analysis. Our results demonstrate the ability of the experimental methodologies by a combination of STEM-ABF, 4D-STEM, and EELS to simultaneously probe local structural and charge information, providing guidance for understanding polarity at the atomic scale in nickelates. These findings also provide inspiration for engineering polarity at the atomic scale in functional materials. For example, surface polarity can be manipulated by modifying the electrostatic properties at the surface with an applied bias voltage, potentially tuning the electrical resistance[46]. In addition, oxygen vacancies can cause strong charge localization with a substantial increase in electrical resistance in nickelates, where the oxygen vacancy distribution can be tuned by changing the polarity of the applied bias electric field to achieve resistance switching[46]. The thickness-dependent oxygen vacancy gradient distribution due to the layer-selective topotactic reduction in our sample is likely to provide guidance for the study of specific functional properties. For example, the oxygen vacancy gradient can be intentionally controlled

to form a *p-n* junction with variations in electrical conductivity in certain oxide materials, which may be useful in devices such as solid oxide fuel cells[47]. In addition, the approach of tuning the surface structure in our work can be applied to explore the modification of surface catalysts, since the structural distortion of the oxygen octahedron strongly correlates with the catalytic activity for the oxygen evolution reaction[48].

## Methods

### Materials
The pristine $Pr_{0.8}Sr_{0.2}NiO_3$ film was grown on a $NdGaO_3$ (110) single crystal of size $1 \times 1$ cm$^2$ using ozone-assisted layer-by-layer molecular beam epitaxy (MBE) with a substrate temperature of ~600 °C and a pressure of $2.4 \times 10^{-7}$ bar under ozone flow. The sample were cut into four pieces of size $5 \times 5$ mm$^2$ and then two of them were reduced by a topochemical reduction process involving heating the samples to 230 °C for 6 and 18 h with 0.1 g $CaH_2$ powder. The sample and powder are separated by aluminum foil in a vacuum-sealed glass tube.

### TEM measurements
The TEM samples were prepared by using a focused ion beam in a high vacuum and further cleaned by a Fischione NanoMill system. STEM imaging and EELS spectrum imaging were carried out using a probe-corrected electron microscope (JEOL ARM200F, JEOL Co. Ltd.) at 200 kV. The STEM-ABF and HAADF images were acquired with collection semi-angles in ranges of 10–20 mrad and 70–300 mrad, respectively. A Gatan K2 camera enables an energy resolution of ~1 eV at a dispersion of 0.5 eV per channel. The collection semi-angle for EELS was 85 mrad. A principle component analysis (PCA) method[49] was used for spectrum image denoising. 4D-STEM experiments were performed in the 1-bit mode with continuous reading and writing at a frame time of $4.8 \times 10^{-5}$ s by using a Merlin direct electron detector ($256 \times 256$ pixels, Quantum Detectors). The 4D-STEM results were optimized using multi-frames acquisition. The electrostatic field maps were calculated by a simplified quantum mechanical model[36,37,50]

$$\mathbf{E}_\perp = -\frac{v}{e}\frac{\Delta\mathbf{p}}{\Delta z} \quad (2)$$

where $\mathbf{E}_\perp$ is the projected electric field in beam direction, which consists of x and y components of the lateral electric field. By converting the shift of the center-of-mass (ΔCoM) to momentum transfer, we can determine the electric field information due to the proportional relationship between the electric field and the change of the momentum transfer of the electron beam ($\Delta\mathbf{p}_\perp$). $\Delta z$ is the sample thickness, $e$ is the elementary charge, and $v$ is the electron velocity.

## Data availability
The data that support the findings of this study are available from the corresponding author upon reasonable request.

## Code availability
The code used in this study is available from the corresponding author on reasonable request.

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

## Acknowledgements

We acknowledge the funding support from the European Union's Horizon 2020 research and innovation programme under Grant Agreement No. 823717 – ESTEEM3. We are thankful to Dr. Y. Wang for the support of 4D-STEM data processing, Dr. T. Heil for the support of Merlin software, U. Salzberger, Dr. J. Deuschle, and M. Kelsch for TEM sample preparation, and TEM support from K. Hahn and P. Kopold.

## Author contributions

C.Y. designed and performed the TEM experiments. W.S., H.G.W., and P.V.A. provide insightful discussions on the TEM results. C.Y. performed the simulation and calculation of TEM images with important discussions with W.S., H.G.W., and P.V.A. R.P. synthesized the nickelate films and measured the electrical properties under the supervision of E.B., G.L., and B.K. R.P. and E.B. provide important discussions on the electrical properties of nickelates. The manuscript was written by C.Y. with contributions from all authors.

## Funding

## Competing interests

The authors declare no competing interests.
