## [Peer Review File · Nature Communications]

Direct observation of strong surface reconstruction in partially reduced nickelate filmsREVIEWER COMMENTS

Reviewer #1 (Remarks to the Author):

This manuscript discusses the structural properties, especially polarization, of $\text{Pr}_{0.8}\text{Sr}_{0.2}\text{NiO}_2$ films grown epitaxially by MBE. The atomic-resolution electron microscope is used to carefully observe the displacement of atoms and the electronic state of the film surfaces. I understand oxides are a fascinating group of materials that exhibit diverse physical properties such as ferroelectricity, superconductivity, and magnetism. However, it is not clearly indicated why the authors focus on the polarization of $\text{Pr}_{0.8}\text{Sr}_{0.2}\text{NiO}_2$ thin film surfaces in the manuscript. Therefore, it is unclear whether the findings extracted from the evaluation of the $\text{Pr}_{0.8}\text{Sr}_{0.2}\text{NiO}_2$ surface can be applied to the entire oxide or compound single-crystal materials. In addition, the authors use only transmission electron microscopy to characterize physical properties and do not present the results of the evaluation of electrical properties (e.g., surface electrical conductivity). While I appreciate that the authors used advanced electron microscopy techniques to discover a new phenomenon in $\text{Pr}_{0.8}\text{Sr}_{0.2}\text{NiO}_2$, it is difficult to believe that the results have the impact that Nature Communications readers would expect. Therefore, I believe that it is appropriate for this paper to undergo another peer review in a journal dedicated to materials science.

Reviewer #2 (Remarks to the Author):

Based on powerful heterogeneous interface growth and characterization techniques, the author conducted a detailed comparative study of the characteristics of $\text{Pr}_{0.8}\text{Sr}_{0.2}\text{NiO}_3$ and $\text{Pr}_{0.8}\text{Sr}_{0.2}\text{NiO}_{2+x}$ thin films from the perspectives of atomic and electronic structures. It was found that surface polarization and reconstruction, oxygen octahedral distortion, and the presence of oxygen vacancies are important, providing a possibility for the subsequent implementation of potential superconducting and related functional properties. The data and conclusions appear reliable and reasonable. But before this article was published, I had the following questions:

1. As mentioned by the author in the introduction, superconductivity has recently been discovered in infinite layer divalent nickelate materials. What kind of inspiration and significance does the author's research in this article have for the research in the field of superconductivity? It can be extended to discuss.

2. The author's discussion on structural characterization has a slightly higher proportion, and the discussion on electronic structure or other physical properties can be more focused and strengthened.

Reviewer #3 (Remarks to the Author):

In this study, Yang et al. used annular bright-field imaging to investigate the surface structure of $\text{Pr}_{0.8}\text{Sr}_{0.2}\text{NiO}_{2+x}$ ($0 < x < 1$) thin films. They observed pronounced polar distortions coupled with octahedral rotations in fully oxidized $\text{Pr}_{0.8}\text{Sr}_{0.2}\text{NiO}_3$ samples. Interestingly, even stronger polar distortions were observed in partially reduced samples. This study is of great significance for effectively enhancing the electrocatalytic and photocatalytic

performance by regulating the adsorption strength and charge separation at the catalyst surface. I suggest accepting it after making some revisions.

1. This study is very interesting, and I am curious if the authors can provide the distribution of the modified crystal field or the relevant filling capacity of molecular orbitals. I noticed that the authors often discuss band tuning using distortions.
2. Is this transport property related to the magnetic properties? The different spin states of magnetic electrons in the d orbitals will greatly affect the electron transfer capability.
3. What is the underlying cause of this physical phenomenon resulting from distortions? Can theoretical chemistry or quantum mechanics be employed to further analyze this aspect?
4. Is there a limit to this tuning method? In other words, is it possible to establish a mapping relationship?
5. It would be helpful if the authors could provide some electronic structure calculations to fully integrate this information, which would facilitate theoretical explanations.
6. Will the hybrid orbitals change with the modulation of distortions, and if so, how do they change? This would be a point of particular interest.
7. In which fields and applications will the emergence of this work contribute to rapid progress? These aspects need further elaboration from the authors.
8. There are some issues with the paragraph organization by the authors, with some paragraphs being excessively long, which decreases readability. I suggest splitting and condensing these paragraphs to improve readability.
9. The enlarged high-resolution annular bright-field images at the (100) surface regions in $\text{Pr}_{0.8}\text{Sr}_{0.2}\text{NiO}_3$ and $\text{Pr}_{0.8}\text{Sr}_{0.2}\text{NiO}_{2+x}$ films should be selected at the same depth to analyze the polar distortion of $\text{Pr}_{0.8}\text{Sr}_{0.2}\text{NiO}_{2+x}$.
10. Figure 2(e) quantitatively compares the relative displacements of Ni and O columns at the NiO_2 plane in the out-of-plane direction between the pristine and reduced samples. However, as described in Figure 2(a) and 2(c), there is no Ni atom below the red dotted line. Therefore, the relative displacements of Ni and O columns below the red dotted line are meaningless in Figure 2(e). Moreover, more areas perpendicular to the red dotted line should be selected to test the displacement of Ni and O columns to eliminate random errors.
11. Figure 4(c) shows a gradual decrease in the maximum intensity ratio of peaks A and B of the O-K edge, indicating a decrease in the hybridization of Ni 3d and O 2p, indicating decreased metallicity. However, more studies reported reduced oxides performed high conductivity, namely improved metallicity. Why reduced $\text{Pr}_{0.8}\text{Sr}_{0.2}\text{NiO}_{2+x}$ film performed decreased hybridization and metallicity? Authors should supplement electronic transport properties to relate metallicity with hybridization rather than discuss in expectation.

Response to the Reviewers' Comments

Reviewer #1 (Remarks to the Author):

This manuscript discusses the structural properties, especially polarization, of $\text{Pr}_{0.8}\text{Sr}_{0.2}\text{NiO}_2$ films grown epitaxially by MBE. The atomic-resolution electron microscope is used to carefully observe the displacement of atoms and the electronic state of the film surfaces. I understand oxides are a fascinating group of materials that exhibit diverse physical properties such as ferroelectricity, superconductivity, and magnetism.

1. However, it is not clearly indicated why the authors focus on the polarization of $\text{Pr}_{0.8}\text{Sr}_{0.2}\text{NiO}_2$ thin film surfaces in the manuscript. Therefore, it is unclear whether the findings extracted from the evaluation of the $\text{Pr}_{0.8}\text{Sr}_{0.2}\text{NiO}_2$ surface can be applied to the entire oxide or compound single-crystal materials.

Response: Thank you for the reviewer's comment. Reconstruction at surfaces or interfaces can modify the physical and chemical properties of complex oxides, such as orbital polarization at LaNiO_3 surface¹, 2DEG at SrTiO_3 surface^{3,4}, charge density wave at KTaO_3 surface⁵, which is expected to influence the behavior of macroscopic material properties, such as superconductivity and magnetism. The recent observation of superconductivity in (A, Sr)NiO₂ (A: La, Nd, Pr) films has aroused considerable interest in infinite-layer nickelates. Some efforts have been made to understand its mechanism. For example, theoretical calculations predict the formation of 2DEG at the $\text{NdNiO}_2/\text{SrTiO}_3$ interface due to strong polarity mismatch. The experimental observation proved the interface reconstruction, although interfacial elemental mixing and residual oxygen reduce the interface polarity. The surface reconstruction is expected to be stronger than the interfacial reconstruction. In addition, the controversial results of the charge density wave state in the infinite layer nickelates associated with the capping layer^{6,7,8,9} may indicate the potentially critical influence of polarity on the surface or interface electronic state. By using a SrTiO_3 capping layer, the polarity at the interface layer will be greatly reduced due to the residual oxygen caused by the elemental mixing at the interface. In our work, we clearly elaborate the surface polarity induced strong reconstruction in nickelates at the atomic scale using advanced STEM techniques, where the surface reconstruction can also be tuned by layer-selective topotactic reduction. Furthermore, previous studies of surface and interface reconstructions in the reduced nickelates lack detailed atomic observations. Although our observations are based only on the partially reduced nickelates, a similar surface reconstruction is expected to be stronger and have an important effect in the infinite layer nickelates.

Our results may also provide valuable insight into the engineering of the surface polarity in the nickelates for resistance switching applications¹⁰. The polar distortion at the surface reduces the overlap of Ni 3d and O 2p orbitals, which affects the electronic behavior. The surface polarity can be manipulated by modifying the electrostatic properties at the surface with an applied bias voltage, potentially tuning the electrical resistance. In addition, oxygen vacancies can cause strong charge localization with a substantial increase in electrical resistance in nickelates, where the oxygen vacancy distribution can be tuned by changing the polarity of the applied electric field to resistance switching¹⁰. The thickness-dependent oxygen vacancy distribution due to the layer-selective topotactic reduction in our sample is likely to provide critical insights into the study of interface and surface devices. In addition, the approach to tune the surface structure in our work can be applied to explore the modification of the surface catalyst, since the structural distortion of the oxygen octahedron modifies the density of states of the O 2p and Ni 3d orbitals, which is strongly correlated with the catalytic activity for the oxygen evolution reaction¹¹. Therefore, we believe that studying the exploitation and

control of surface polarization and reconstruction at the atomic scale can advance the understanding and engineering of polarity in functional materials.

2. In addition, the authors use only transmission electron microscopy to characterize physical properties and do not present the results of the evaluation of electrical properties (e.g., surface electrical conductivity). While I appreciate that the authors used advanced electron microscopy techniques to discover a new phenomenon in $\text{Pr}_{0.8}\text{Sr}_{0.2}\text{NiO}_2$, it is difficult to believe that the results have the impact that Nature Communications readers would expect. Therefore, I believe that it is appropriate for this paper to undergo another peer review in a journal dedicated to materials science.

Response: Thank you for your feedback and for taking the time to review our paper. We acknowledge your point regarding the characterization methods used in our study. Indeed, we focused primarily on electron microscopy techniques to characterize the physical properties of the reduced nickelates. While our paper does not include an evaluation of electrical properties, we believe that the study of the exploitation and control of surface polarization and reconstruction at the atomic scale, as revealed by advanced electron microscopy techniques, is significant for the understanding and engineering of polarity in functional materials, which can serve as a foundation for further investigations.

Reviewer #2 (Remarks to the Author):

Based on powerful heterogeneous interface growth and characterization techniques, the author conducted a detailed comparative study of the characteristics of $\text{Pr}_{0.8}\text{Sr}_{0.2}\text{NiO}_3$ and $\text{Pr}_{0.8}\text{Sr}_{0.2}\text{NiO}_{2+x}$ thin films from the perspectives of atomic and electronic structures. It was found that surface polarization and reconstruction, oxygen octahedral distortion, and the presence of oxygen vacancies are important, providing a possibility for the subsequent implementation of potential superconducting and related functional properties. The data and conclusions appear reliable and reasonable. But before this article was published, I had the following questions:

1. As mentioned by the author in the introduction, superconductivity has recently been discovered in infinite layer divalent nickelate materials. What kind of inspiration and significance does the author's research in this article have for the research in the field of superconductivity? It can be extended to discuss.

Response: Thank you for this valuable suggestion. We have added further discussion of the inspiration and significance of our work in the nickelate superconductor system on pages 2-3, 10. Reconstruction at surfaces or interfaces can modify the physical and chemical properties of complex oxides, such as orbital polarization^{1,2}, 2DEG^{3,4}, charge density wave⁵, which are expected to influence the response of macroscopic material properties. For example, theoretical calculations predict the formation of 2DEG at the $\text{NdNiO}_2/\text{SrTiO}_3$ interface due to the strong polarity mismatch, which is similar to the $\text{LaAlO}_3/\text{SrTiO}_3$ superconducting system. However, the experimental results show that the residual oxygen, elemental mixing, atomic reconstruction, and Ni valence modulations can compensate for the polar discontinuity induced by the interface polarity at the practical $\text{NdNiO}_2/\text{SrTiO}_3$ interface. In terms of surface polarity, theoretical calculations predict a high Ni e_g orbital polarization of 35% due to the elimination of out-of-plane Ni-O bonding, which is associated with polar distortion. However, the experimental study of the effects of the surface property is lacking due to the difficulty in sample synthesis and characterization. Recently, the controversial results of the charge density wave state in the infinite layer nickelates associated with the capping layer^{6,7,8,9} may indicate the potentially critical influence of polarity on the surface or interface electronic state. By using a SrTiO_3 capping layer, the surface polarity is greatly reduced due to elemental mixing and residual oxygen. In addition, the homogeneity of oxygen deintercalation in the bulk region can also affect the geometry of the infinite layer phase. Understanding the reduction process is crucial for better control of this process. Therefore, detailed studies of interfacial, surface, and geometry effects at the atomic scale are essential to understand, exploit, and control oxygen deintercalation in nickelate superconductors and similar systems. In our work, we have clearly elaborated the surface polarity induced strong reconstruction in nickelates at the atomic scale using advanced STEM techniques, discussed its electronic structure implications, and directly imaged the electrostatic field distribution and tunable surface reconstruction. We also demonstrate thickness-dependent oxygen deintercalation by layer-selective topotactic reduction.

2. The author's discussion on structural characterization has a slightly higher proportion, and the discussion on electronic structure or other physical properties can be more focused and strengthened.

Response: Thank you for these very constructive suggestions. First, we have added more discussion on the relationship between atomic structure and Ni $3d$ -O $2p$ orbital hybridization in the nickelate system on page 7 of the manuscript. As shown schematically in Figure R1, crystal field splitting occurs due to the ligand oxygen in the NiO_5 pyramidal structure¹⁰. A Ni-O displacement reduces the Ni-O-Ni angle, which reduces the hybridization between the Ni $3d_{x^2-y^2}$ and O $2p_x$ orbitals. There is a canonical tight binding approach to estimate the changes in the bond overlap in nickelates, which gives a transfer integral t_{pd} between Ni $3d$ and O $2p$ orbitals^{12,13}:

$$t_{pd} = k \frac{\cos(90 - \theta/2)}{d^{3.5}}$$

where k is a constant determined by the p - d orbital covalent hopping integral, θ is the in-plane Ni-O-Ni bond angle, and d is the Ni-O bond length. Larger values of t_{pd} imply a greater tendency toward metallic behavior in the nickelate system². Smaller t_{pd} values in the surface layer than in the bulk region indicate reduced Ni 3*d* and O 2*p* hybridization.

Figure R1 Schematic diagram of the crystal field splitting of the Ni 3*d* orbitals and the Ni 3*d*-O 2*p* orbital overlap in (a) a NiO₅ pyramidal structure and (b) a polar distorted NiO₅ pyramidal structure.

In addition, we have added the results of the effect of the structural change on the electronic structure at the nickelate surface according to the DFT calculations on pages 7-8 of the manuscript. We have included the detailed description of the figures in the Supplementary Information. Figure R2 (a) shows the PrNiO₃ supercell transformed from the PrNiO₃ orthorhombic unit cell. A NiO₂ terminated surface was created and a 10 Å vacuum layer was added. Figure R2 (b) shows the projected density of states of the O 2*p* and Ni 3*d* orbitals where they hybridize near the Fermi level. Figure R2 (c) shows the PrNiO₃ supercell with a clear surface polar distortion modified according to the experimental result, where there is a Ni-O-Ni buckling structure and the apical oxygen moves away from the Ni atoms due to the electrostatic field at the surface. The Ni-O displacement is set to 0.5 Å, similar to the result in Figure 2(e). The corresponding electronic states decrease near the Fermi level marked by the orange shadow in Figure R2 (d), compared to the state of the undistorted structure in Figure R2 (b). After structure relaxation, there is a slight change in the surface structure in Figure R2 (e) and the corresponding DOS plot in Figure R2 (f) is similar. In addition, we introduce the apical oxygen vacancy at the surface layer, forming an infinite layer structure. After structure optimization, we find that there is an obvious decrease in the out-of-plane Ni-Ni distance and a Ni-O-Ni buckling structure in Figure R2(g), which is consistent with the reported result in a similar NdNiO₂ system¹⁴. From the projected DOS plot in Figure R2(h), we can see the further decreased orbital overlap between the Ni 3*d* and O 2*p* near the Fermi level, marked by the orange shadow. This reduces the charge transfer capability between Ni and oxygen.

Figure R2 Structure change and corresponding electronic density of states (DOS) calculated by DFT. (a) Unrelaxed PrNiO₃ supercell with a [NiO₂] surface, (c) unrelaxed PrNiO₃ supercell with a distorted [NiO₂] surface modified according to the experimental result, (e) relaxed PrNiO₃ supercell in (c), and (g) relaxed PrNiO₂ supercell with a [NiO₂] surface. All structure models have a 10 Å vacuum layer. The corresponding projected DOS plots are shown in (b), (d), (f), and (h), respectively.

Reviewer #3 (Remarks to the Author):

In this study, Yang et al. used annular bright-field imaging to investigate the surface structure of $\text{Pr}_{0.8}\text{Sr}_{0.2}\text{NiO}_{2+x}$ ($0 < x < 1$) thin films. They observed pronounced polar distortions coupled with octahedral rotations in fully oxidized $\text{Pr}_{0.8}\text{Sr}_{0.2}\text{NiO}_3$ samples. Interestingly, even stronger polar distortions were observed in partially reduced samples. This study is of great significance for effectively enhancing the electrocatalytic and photocatalytic performance by regulating the adsorption strength and charge separation at the catalyst surface. I suggest accepting it after making some revisions.

1. This study is very interesting, and I am curious if the authors can provide the distribution of the modified crystal field or the relevant filling capacity of molecular orbitals. I noticed that the authors often discuss band tuning using distortions.

Response: Thank you for this valuable question. We have added a more detailed discussion of the modified crystal field on page 8 of the manuscript. As shown schematically in Figure R1, crystal field splitting occurs due to the ligand oxygen in the NiO_5 pyramidal structure¹⁰. A Ni-O displacement reduces the Ni-O-Ni angle, which reduces the hybridization between the Ni $3d_{x^2-y^2}$ and O $2p_x$ orbitals. There is a canonical tight binding approach to estimate the changes in bond overlap in nickelates, which gives a transfer integral t_{pd} between Ni $3d$ and O $2p$ orbitals^{12, 13}:

$$t_{pd} = k \frac{\cos(90 - \theta/2)}{d^{3.5}}$$

where k is a constant determined by the p - d orbital covalent hopping integral, θ is the in-plane Ni-O-Ni bond angle, and d is the Ni-O bond length. Larger values of t_{pd} imply a greater tendency toward metallic behavior in the nickelate system¹². Smaller t_{pd} values in the surface layer than that in the bulk region indicate a reduced Ni $3d$ and O $2p$ hybridization.

Figure R1 Schematic diagram of crystal field splitting of Ni $3d$ orbitals and the Ni $3d$ -O $2p$ orbital overlap in (a) a NiO_5 pyramid structure and (b) a polar distorted NiO_5 pyramid structure.

2. Is this transport property related to the magnetic properties? The different spin states of magnetic electrons in the d orbitals will greatly affect the electron transfer capability.

Response: Thank you for the very thoughtful question. A systematic study of the transport and magnetic properties of reduced $\text{Pr}_{0.8}\text{Sr}_{0.2}\text{NiO}_{2+x}$ was not the goal of the present study and is indeed not an easy task. In a related study of the effects of oxygen reduction on the magnetic properties of SmNiO_{3-x} by Li *et al.*¹⁵, oxygen vacancies were systematically created by heating in ultra-high vacuum. In this way, electron doping levels of 0.835 per Ni could be stabilized. Below the Neel temperature (T_{Neel}), fully oxygenated bulk SmNiO_3 exhibits a non-collinear antiferromagnetic order with a characteristic $(1/4, 1/4, 1/4)_{\text{pc}}$ ordering vector in pseudo-cubic notation¹⁵. The same order has been reported for PrNiO_3 , although with different $T_{\text{Neel}} \sim 120$ K. In the study by Li *et al.* resonant elastic x-ray scattering was used to study the magnetic reflection and its disappearance above doping levels of 0.21 electrons doped per Ni. The explanation for this observation is that oxygen vacancies are responsible for the disruption of the superexchange interaction network.

The samples investigated in the present study differ from the SmNiO_x films of Li *et al.* in two aspects: First, the cation doping with Sr in the present samples has an effect on the electronic and magnetic structure even in non-reduced samples. Second, the reduction by reaction with CaH_2 results in significantly more reduced phases, up to $\text{RNiO}_{2.0}$, than heating in vacuum. Although we have not been able to quantify the exact oxidation state of our thin films, based on the XRD characterization we assume that we have more reduced films. Thus, we assume that the bulk-like AF order is not present in the films studied here. However, other types of magnetic order cannot be excluded. As mentioned above, the study of complex magnetic order will have to be a central goal of future studies.

3. What is the underlying cause of this physical phenomenon resulting from distortions? Can theoretical chemistry or quantum mechanics be employed to further analyze this aspect?

Response: We appreciate the reviewer's valuable question. Polar distortion is a kind of atomic reconstruction at the surface due to the polar discontinuity induced by the presence of electric dipoles or charge imbalances at the surface. From a layer-by-layer perspective, the surface consisting of the negatively charged $[\text{NiO}_2]^-$ layer and the vacuum induces the polar discontinuity, resulting in a built-in electric field. In the case of the reduced samples such as the infinite layer nickelates, the polar instability becomes stronger due to a larger charge imbalance between the negatively charged $[\text{NiO}_2]^{3-}$ layer and the vacuum. To avoid polar instability, electronic and atomic reconstructions will occur¹⁶. According to the self-consistent charge-transfer model in the DFT calculation¹⁴, the electronic reconstruction can only partially screen the built-in electric field by charge transfer between Ni sites in adjacent layers while retaining some residual electrostatic energy and leading to the atomic reconstruction. For example, a NiO_5 pyramidal structure at the $\text{NdNiO}_2/\text{SrTiO}_3$ interface shows a ~ 0.2 Å displacement of Ni in the NiO_2 layer¹⁴. This is in agreement with our previous experimental results measured by STEM at the interfaces in the $8\text{NdNiO}_2/4\text{SrTiO}_3$ superlattice¹⁷. In the case of the partially reduced sample in this manuscript, a similar NiO_5 pyramidal structure was formed at the top surface layer. In addition, the polar distorted NiO_6 structure was formed at the subsurface layers to compensate for the built-in electric field and gradually decayed in about 3 unit cells. In addition, we observed that the direction of the polar distortion is opposite to that of the built-in electric field. We have included the discussion for comparison with the previous literature on page 9 of the manuscript.

4. Is there a limit to this tuning method? In other words, is it possible to establish a mapping relationship?

Response: We thank the reviewer for the very valuable question. The limitations of the tuning method of topological reduction include the stability of the reduced structure and the uniformity of the oxygen

deintercalation. For example, a challenging experiment in this field so far is the synthesis of superconducting infinite layer nickelates, because a Ruddlesden-Popper-type secondary phase and film decomposition easily occur during the reduction process^{18, 19}. The infinite layer nickelates can only be stabilized below 10 nm with a capping layer on the surface²⁰. Without the capping layer, the thickness of the stabilized infinite layer structure is much thinner with a few nm, where the exact value has not been clearly claimed. With the current state of research on topotactically reduced PSNO thin films, we know that different final states are possible (two of which are presented in this paper), but determining and accurately tuning the final oxygen concentration is difficult. In the case of tuning the reduction process to synthesize the superconducting infinite layer phase, unfortunately, we are not yet able to give a clear mapping relationship so far, which requires further efforts in this field.

In addition, the surface modification by topological reduction is relatively easier to control than the synthesis of the superconducting infinite layer phase. For example, we compared the surface structures of the partially reduced samples with different reduction time, as the HAADF images, as shown in Figure S2. A small amount of structural decomposition starts to form in the surface region in the sample with more reduction time. The subsurface reconstruction was quantified by carefully applying the advanced TEM sample preparation and STEM imaging techniques. However, further reduction strength may induce more amorphous phase in the surface region, making it unavailable for the detailed studies at the atomic scale. Another limitation of the topological reduction method is that the oxygen deintercalation is not homogeneous in the film growth direction. As shown in Figure 5f and our previously published results in the $4\text{NdNiO}_2/2\text{SrTiO}_3$ superlattice¹⁷, the oxygen deintercalation is thickness dependent. That is, the greater the thickness of the nickelate reduction, the greater the risk of decomposition of the near-surface infinite layer phase.

5. It would be helpful if the authors could provide some electronic structure calculations to fully integrate this information, which would facilitate theoretical explanations.

Response: We appreciate the very constructive suggestion of the reviewer. It is well known that structural distortion can be strongly correlated with changes in the energy state of electrons in transition metal oxides. We have added the results of the effect of the structural change on the electronic structure at the nickelate surface on page 8. We have included the detailed description of the figures in the Supplementary Information. Figure R2 (a) shows the PrNiO_3 supercell transformed from the PrNiO_3 orthorhombic unit cell. A NiO_2 terminated surface was created and a 10 Å vacuum layer was added. Figure R2 (b) shows the projected density of states of the O 2*p* and Ni 3*d* orbitals where they hybridize near the Fermi level. Figure R2 (c) shows the PrNiO_3 supercell with a clear surface polar distortion modified according to the experimental result, where there is a Ni-O-Ni buckling structure and the apical oxygen moves away from the Ni atoms due to the electrostatic field at the surface. The Ni-O displacement is set to 0.5 Å, similar to the result in Figure 2(e). The corresponding electronic states decrease near the Fermi level marked by the orange shadow in Figure R2 (d), compared to the state of the undistorted structure in Figure R2 (b). After structure relaxation, there is a slight change in the surface structure in Figure R2 (e) and the corresponding DOS plot in Figure R2 (f) is similar. In addition, we introduce the apical oxygen vacancy at the surface layer, forming an infinite layer structure. After structure optimization, we find that there is an obvious decrease in the out-of-plane Ni-Ni distance and a Ni-O-Ni buckling structure in Figure R2(g), which is consistent with the reported result in a similar NdNiO_2 system¹⁴. From the projected DOS plot in Figure R2(h), we can see the further decreased orbital overlap between the Ni 3*d* and O 2*p* near the Fermi level, marked by the orange shadow. This reduces the charge transfer capability between Ni and oxygen.

Figure R2 Structure change and corresponding electronic density of states (DOS) calculated by DFT. (a) Unrelaxed PrNiO₃ supercell with a [NiO₂] surface, (c) unrelaxed PrNiO₃ supercell with a distorted [NiO₂] surface modified according to the experimental result, (e) relaxed PrNiO₃ supercell in (c), and (g) relaxed PrNiO₂ supercell with a [NiO₂] surface. All structure models have a 10 Å vacuum layer. The corresponding projected DOS plots are shown in (b), (d), (f), and (h), respectively.

6. Will the hybrid orbitals change with the modulation of distortions, and if so, how do they change? This would be a point of particular interest.

Response: Thank you for this question. There is a canonical tight binding approach to estimate the changes in bond overlap that gives a transfer integral t_{pd} between Ni 3d and O 2p orbitals^{12, 13}:

$$t_{pd} = k \frac{\cos(90 - \theta/2)}{d^{3.5}}$$

where k is a constant determined by the p - d orbital covalent hopping integral, θ is the in-plane Ni-O-Ni bond angle, and d is the Ni-O bond length. A strong polar distortion in our sample reduces the Ni-O-Ni angle in the surface region, which reduces the overlap or hybridization between the Ni $3d_{x^2-y^2}$ and O 2p orbitals (see Figure R1) and leads to a reduction in the valence bandwidth^{12, 21, 22, 23}. The reduced bandwidth can lead to the opening of the charge transfer gap between the Ni e_g and O p valence bands^{12, 24}. We have added this discussion to the manuscript on page 8.

7. In which fields and applications will the emergence of this work contribute to rapid progress? These aspects need further elaboration from the authors.

Response: Thank you for this thoughtful question. In our work, we clearly elaborate the surface polarity induced strong reconstruction in nickelates at the atomic scale using advanced STEM techniques, where the surface reconstruction can be tuned by layer-selective topotactic reduction. Surfaces or interfacial reconstruction can modify the physical and chemical properties of complex oxides, such as orbital polarization^{1, 2}, 2DEG^{3, 4}, charge density wave⁵, which are expected to influence the response of macroscopic material properties. Recently, the controversial results of the charge density wave state in the infinite layer nickelates associated with the capping layer^{6, 7, 8, 9} may indicate the potentially critical influence of polarity on the surface or interfacial electronic state. Although our observations are based only on the partially reduced nickelates, a similar surface reconstruction is expected to be stronger and have an important effect in the infinite layer nickelates.

Our results may also provide valuable insight into the engineering of the surface polarity in the nickelates for resistance switching applications¹⁰. The polar distortion at the surface reduces the overlap of the Ni 3*d* and O 2*p* orbitals, resulting in a reduction in conductivity. The surface polarity can be manipulated by modifying the electrostatic properties at the surface with an applied bias voltage, potentially tuning the electrical resistance. In addition, oxygen vacancies can cause strong charge localization with a substantial increase in electrical resistance in nickelates, where the oxygen vacancy distribution can be tuned by changing the polarity of the applied electric field to resistive switching¹⁰. The thickness-dependent oxygen vacancy distribution due to the layer-selective topotactic reduction in our sample is likely to provide critical insights into the study of interface and surface devices. In addition, the approach of tuning the surface structure can be applied to explore the modification of surface catalysts, since the structural distortion of the oxygen octahedron is strongly correlated with the catalytic activity for the oxygen evolution reaction¹¹. In our work, we directly image the cooperative coupling of polar distortion and octahedral rotation in the pristine sample, and a strong surface reconstruction in the reduced nickelates, which has been overlooked in recent studies. The experimental methodology demonstrated in this work allows the simultaneous measurement of the atomic structure and the electrostatic charge distribution in the surface region in detail, thus greatly advancing the understanding and engineering of polarity at the atomic scale in functional materials. We have added these discussions to the manuscript on pages 2 and 10.

8. There are some issues with the paragraph organization by the authors, with some paragraphs being excessively long, which decreases readability. I suggest splitting and condensing these paragraphs to improve readability.

Response: Thank you for pointing out this important detail. We have split the paragraph on page 5 regarding the discussion of the 4D-STEM simulation results. We have modified the discussion part on pages 7-8 and rearranged the paragraph to improve readability accordingly.

9. The enlarged high-resolution annular bright-field images at the (100) surface regions in Pr_{0.8}Sr_{0.2}NiO₃ and Pr_{0.8}Sr_{0.2}NiO_{2+x} films should be selected at the same depth to analyze the polar distortion of Pr_{0.8}Sr_{0.2}NiO_{2+x}.

Response: Thank you for pointing out this detail. We have changed Figure 2 in the manuscript accordingly.

10. Figure 2(e) quantitatively compares the relative displacements of Ni and O columns at the NiO₂ plane in the out-of-plane direction between the pristine and reduced samples. However, as described in Figure 2(a) and 2(c), there is no Ni atom below the red dotted line. Therefore, the relative displacements

of Ni and O columns below the red dotted line are meaningless in Figure 2(e). Moreover, more areas perpendicular to the red dotted line should be selected to test the displacement of Ni and O columns to eliminate random errors.

Response: Thank you for this valuable suggestion. We have modified the Figure 2 on page 13. The following Fig. R3 shows the quantification of the displacement of Ni and O columns from more selected areas perpendicular to the red dotted line.

Figure R3: Oxygen sublattices in Pr_{0.8}Sr_{0.2}NiO₃ and Pr_{0.8}Sr_{0.2}NiO_{2+x} films. ABF images of (a) Pr_{0.8}Sr_{0.2}NiO₃ and (c) Pr_{0.8}Sr_{0.2}NiO_{2+x} films. (b) Enlarged ABF images and the corresponding schematic structural models (right) show the variation of oxygen octahedra in the pristine sample. (d) Enlarged ABF images and corresponding schematic structural models (right) show the variation of oxygen octahedra in the reduced sample. (e) The relative displacement of Ni and O at the NiO₂ plane in the out-of-plane direction. The schematic atomic structure at the PrNiO₃/NdGaO₃ interface.

11. Figure 4(c) shows a gradual decrease in the maximum intensity ratio of peaks A and B of the O-K edge, indicating a decrease in the hybridization of Ni 3d and O 2p, indicating decreased metallicity. However, more studies reported reduced oxides performed high conductivity, namely improved metallicity. Why reduced Pr_{0.8}Sr_{0.2}NiO_{2+x} film performed decreased hybridization and metallicity? Authors should supplement electronic transport properties to relate metallicity with hybridization rather than discuss in expectation.

Response: We appreciate the reviewer's question. We have corrected the inaccurate description on page 7 of the manuscript. In rare earth nickelates, as the ionic radius of the rare earth decreases (La to Eu), the Ni-O-Ni angle decreases and prevents the overlap of the Ni 3d and O 2p orbitals, leading to a decrease in electrical conductivity.¹³ However, oxygen vacancies can also lead to carrier localization in nickelates and lead to a decrease in electrical conductivity.¹⁰ The O-K edge reflects the site- and symmetry-projected unoccupied DOS and not the total DOS. Therefore, it does not necessarily indicate reduced metallicity. In fact, XAS above and below the metal-insulator transition (three orders of magnitude change in resistivity) show little change in the O-K pre-peak intensity (peak A) (see e.g. Fig. 2 B in Ref.²⁵). Rather, the prepeak is a signature of the complex band structure of the rare-earth nickelates and results from their negative charge-transfer nature²⁶. Upon reduction to the infinite-layer nickelate phase, XAS and EELS measurements show that the pre-peak disappears at the O-K edge^{27, 28}. Also, when comparing the actual resistivity values for the first discovered superconducting infinite layer nickelate²⁹, the normal resistivity values of the doped sample and the resistivity of the undoped infinite-layer are higher (by a factor of about 10) than their perovskite precursor.

For the specific samples of reduced Pr_{0.8}Sr_{0.2}NiO_{2+x} from this STEM study, we cannot provide resistance versus temperature curves due to the destructive preparation method. However, the original thin film sample was cut into several pieces prior to reduction and we have reduced another piece under similar

conditions. As presented in the study, it was vacuum sealed in a glass tube containing 0.1 g CaH₂ with the sample and powder separated by aluminum foil. Then the glass tube was heated to 220°C for 12 hours. The reduction time is between the two studied by STEM and discussed in the main text. The corresponding resistivity data are shown below in comparison to the data for the pristine, unreduced perovskite piece. As the graph shows, the reduction results in a semiconducting temperature dependence of the resistivity that has been qualitatively reproduced for other samples.

Figure R4: Temperature-dependent resistivity data for the as-grown, pristine perovskite thin film Pr_{0.8}Sr_{0.2}NiO₃ (black line) and a reduced piece of this sample (red line), measured with a van-der-Pauw 4-point probe in a Quantum Design PPMS.

References.

1. Chen H, Kumah DP, Disa AS, Walker FJ, Ahn CH, Ismail-Beigi S. Modifying the electronic orbitals of nickelate heterostructures via structural distortions. *Phys. Rev. Lett.* **110**, 186402 (2013).
2. Liao Z, *et al.* Large orbital polarization in nickelate-cuprate heterostructures by dimensional control of oxygen coordination. *Nat. Commun.* **10**, 589 (2019).
3. Geisler B, Pentcheva R. Correlated interface electron gas in infinite-layer nickelate versus cuprate films on SrTiO₃(001). *Phys. Rev. Res.* **3**, 013261 (2021).
4. Meevasana W, *et al.* Creation and control of a two-dimensional electron liquid at the bare SrTiO₃ surface. *Nat. Mater.* **10**, 114-118 (2011).
5. Reticcioli M, *et al.* Competing electronic states emerging on polar surfaces. *Nat. Commun.* **13**, 4311 (2022).
6. Tam CC, *et al.* Charge density waves in infinite-layer NdNiO₂ nickelates. *Nat. Mater.* **21**, 1116-1120 (2022).
7. Krieger G, *et al.* Charge and Spin Order Dichotomy in NdNiO₂ Driven by the Capping Layer. *Phys. Rev. Lett.* **129**, 027002 (2022).
8. Rossi M, *et al.* A broken translational symmetry state in an infinite-layer nickelate. *Nat. Phys.* **18**, 869-873 (2022).
9. J. Pellicciari NK, *et al.* Comment on newly found Charge Density Waves in infinite layer Nickelates. *arXiv:230615086* (2023).
10. Kotiuga M, *et al.* Carrier localization in perovskite nickelates from oxygen vacancies. *Proc. Natl. Acad. Sci. U.S.A.* **116**, 21992-21997 (2019).
11. Bak J, Bin Bae H, Chung SY. Atomic-scale perturbation of oxygen octahedra via surface ion exchange in perovskite nickelates boosts water oxidation. *Nat. Commun.* **10**, 2713 (2019).
12. Kumah DP, *et al.* Tuning the structure of nickelates to achieve two-dimensional electron conduction. *Adv. Mater.* **26**, 1935-1940 (2014).
13. Torrance JB, *et al.* Systematic study of insulator-metal transitions in perovskites RNiO₃ (R=Pr,Nd,Sm,Eu) due to closing of charge-transfer gap. *Phys. Rev. B Condens. Matter* **45**, 8209-8212 (1992).
14. He R, *et al.* Polarity-induced electronic and atomic reconstruction at NdNiO₂/SrTiO₃ interfaces. *Phys. Rev. B* **102**, (2020).
15. Li J, *et al.* Sudden Collapse of Magnetic Order in Oxygen-Deficient Nickelate Films. *Phys. Rev. Lett.* **126**, 187602 (2021).
16. Nakagawa N, Hwang HY, Muller DA. Why some interfaces cannot be sharp. *Nat. Mater.* **5**, 204-209 (2006).
17. Yang C, *et al.* Thickness-Dependent Interface Polarity in Infinite-Layer Nickelate Superlattices. *Nano Lett.* **23**, 3291-3297 (2023).
18. Lee K, *et al.* Aspects of the synthesis of thin film superconducting infinite-layer nickelates. *APL Mater.* **8**, 041107 (2020).
19. Yang C, *et al.* Ruddlesden–Popper Faults in NdNiO₃ Thin Films. *Symmetry* **14**, 464 (2022).
20. Wang BY, *et al.* Isotropic Pauli-limited superconductivity in the infinite-layer nickelate Nd_{0.775}Sr_{0.225}NiO₂. *Nat. Phys.* **17**, 473-477 (2021).
21. Barman SR, Chainani A, Sarma DD. Covalency-driven unusual metal-insulator transition in nickelates. *Phys. Rev. B Condens. Matter* **49**, 8475-8478 (1994).
22. Garcia-Munoz JL, Rodriguez-Carvajal J, Lacorre P, Torrance JB. Neutron-diffraction study of RNiO₃ (R=La,Pr,Nd,Sm): Electronically induced structural changes across the metal-insulator transition. *Phys. Rev. B Condens. Matter* **46**, 4414-4425 (1992).
23. Okimoto Y, Katsufuji T, Okada Y, Arima T, Tokura Y. Optical spectra in (La,Y)TiO₃: Variation of Mott-Hubbard gap features with change of electron correlation and band filling. *Phys. Rev. B Condens. Matter* **51**, 9581-9588 (1995).
24. Medarde M. Structural, magnetic and electronic properties of RNiO₃ perovskites (R = rare earth). *J. Condens. Matter Phys.* **9**, , 1679-1707 (1997).
25. Meyers D, *et al.* Pure electronic metal-insulator transition at the interface of complex oxides. *Sci. Rep.* **6**, 27934 (2016).

26. Bisogni V, *et al.* Ground-state oxygen holes and the metal-insulator transition in the negative charge-transfer rare-earth nickelates. *Nat. Commun.* **7**, 13017 (2016).
27. Hepting M, *et al.* Electronic structure of the parent compound of superconducting infinite-layer nickelates. *Nat. Mater.* **19**, 381-385 (2020).
28. Goodge BH, *et al.* Doping evolution of the Mott–Hubbard landscape in infinite-layer nickelates. *Proc. Natl. Acad. Sci. U.S.A.* **118**, e2007683118 (2021).
29. Li D, *et al.* Superconductivity in an infinite-layer nickelate. *Nature* **572**, 624-627 (2019).

REVIEWERS' COMMENTS

Reviewer #1 (Remarks to the Author):

The importance of this study was added to the Introduction to clarify why the $\text{Pr}_{0.8}\text{Sr}_{0.2}\text{NiO}_2$ needs to be studied. As I pointed out in my first comment, the lack of data on electrical properties of $\text{Pr}_{0.8}\text{Sr}_{0.2}\text{NiO}_2$ is a weak point of this manuscript, but it is acknowledged that the results are extremely sophisticated observations made with advanced transmission electron microscopy. Therefore, the revised manuscript can be judged to be appropriate as an article in Nature Communications.

Reviewer #2 (Remarks to the Author):

The author has answered all my questions and the manuscript is now ready for publication.

Reviewer #3 (Remarks to the Author):

The author perfectly answered all my doubts. This is a well organized paper with a concise form and in-depth explanation. I recommend that this paper be published in a high-level journal such as Nature Communications.